# Multi-Criteria Dimensionality Reduction with Applications to Fairness

Uthaipon (Tao) Tantipongpipat[*][†]        Samira Samadi [*][‡]

Mohit Singh[*][†]        Jamie Morgenstern[*]        Santosh Vempala[*][‡]

## Abstract

Dimensionality reduction is a classical technique widely used for data analysis. One foundational instantiation is Principal Component Analysis (PCA), which minimizes the average reconstruction error. In this paper, we introduce the *multi-criteria dimensionality reduction* problem where we are given multiple objectives that need to be optimized simultaneously. As an application, our model captures several fairness criteria for dimensionality reduction such as the Fair-PCA problem introduced by Samadi et al. [2018] and the Nash Social Welfare (NSW) problem. In the Fair-PCA problem, the input data is divided into $k$ groups, and the goal is to find a single d-dimensional representation for all groups for which the maximum reconstruction error of any one group is minimized. In NSW the goal is to maximize the product of the individual variances of the groups achieved by the common low-dimensional space.

Our main result is an exact polynomial-time algorithm for the two-criteria dimensionality reduction problem when the two criteria are increasing concave functions. As an application of this result, we obtain a polynomial time algorithm for Fair-PCA for $k = 2$ groups, resolving an open problem of Samadi et al. [2018], and a polynomial time algorithm for NSW objective for $k = 2$ groups. We also give approximation algorithms for $k > 2$. Our technical contribution in the above results is to prove new low-rank properties of extreme point solutions to semi-definite programs. We conclude with experiments indicating the effectiveness of algorithms based on extreme point solutions of semi-definite programs on several real-world datasets.

## 1   Introduction

Dimensionality reduction is the process of choosing a low-dimensional representation of a large, high-dimensional data set. It is a core primitive for modern machine learning and is being used in image processing, biomedical research, time series analysis, etc. Dimensionality reduction can be used during the preprocessing of the data to reduce the computational burden as well as at the final stages of data analysis to facilitate data summarization and data visualization [Raychaudhuri et al., 1999; Iezzoni and Pritts, 1991]. Among the most ubiquitous and effective of dimensionality reduction techniques in practice are Principal Component Analysis (PCA) [Pearson, 1901; Jolliffe, 1986; Hotelling, 1933], multidimensional scaling [Kruskal, 1964], Isomap [Tenenbaum et al., 2000], locally linear embedding [Roweis and Saul, 2000], and t-SNE [Maaten and Hinton, 2008].

---

[*]Georgia Institute of Technology.     {tao,ssamadi6}@gatech.edu,    mohit.singh@isye.gatech.edu, jamiemmt.cs@gatech.edu, vempala@cc.gatech.edu

[†]Supported by NSF-AF:1910423 and NSF-AF:1717947.

[‡]Supported in part by NSF awards CCF-1563838 and CCF-1717349.

One of the major obstacles to dimensionality reduction tasks in practice is complex high-dimensional data structures that lie on multiple different low-dimensional subspaces. For example, Maaten and Hinton [2008] address this issue for low-dimensional visualization of images of objects from diverse classes seen from various viewpoints, or Samadi et al. [2018] study PCA on human data when different groups in the data (e.g., high-educated vs low-educated or men vs women) have an inherently different structure. Although these two contexts might seem unrelated, our work presents a general framework that addresses both issues. In both setting, a single criteria for the dimensionality reduction might not be sufficient to capture different structures in the data. This motivates our study of multi-criteria dimensionality reduction.

As an illustration, consider applying PCA on a high dimensional data to do a visualization analysis in low dimensions. Standard PCA aims to minimize the single criteria of average reconstruction error over the whole data. But the reconstruction error on different parts of data can be widely different. In particular, Samadi et al. [2018] show that on real world data sets, PCA has more reconstruction error on images of women vs images of men. A similar phenomenon is also noticed on other data sets when groups are formed based on education. Unbalanced average reconstruction error or equivalently unbalanced variance could have implications of representational harms [Crawford, 2017] in early stages of data analysis.

**Multi-criteria dimensionality reduction.** Multi-criteria dimensionality reduction could be used as an umbrella term with specifications changing based on the applications and the metrics that the machine learning researcher has in mind. Aiming for an output with a balanced error over different subgroups seems to be a natural choice as reflected by minimizing the maximum of average reconstruction errors studied by Samadi et al. [2018] and maximizing geometric mean of the variances of the groups, which is the well-studied Nash social welfare (NSW) objective [Kaneko and Nakamura, 1979; Nash Jr, 1950]. Motivated by these settings, the more general question that we would like to study is as following.

**Question 1.** *How might one redefine dimensionality reduction to produce projections which optimize different groups' representation in a balanced way?*

For simplicity of explanation, we first describe our framework for PCA, but the approach is general and applies to a much wider class of dimensionality reduction techniques. Consider the data points as rows of an $m \times n$ matrix $A$. For PCA, the objective is to find an $n \times d$ projection matrix $P$ that maximizes the Frobenius norm, $\|AP\|_F^2$ (this is equivalent to minimizing the reconstruction error). Suppose that the rows of $A$ belong to different *groups*, based on demographics or some other semantically meaningful clustering. The definition of these groups need not be a partition; each group could be defined as a different weighting of the data set (rather than a subset, which is a 0/1 weighting). Multi-criteria dimensionality reduction can then be viewed as simultaneously considering objectives on the different weightings of $A$. One way to balance multiple objectives is to find a projection $P$ that maximizes the minimum objective value over each of the groups (weightings), i.e.,

$$\max_{P:P^T P=I_d} \min_{1\leq i \leq k} \|A_i P\|_F^2 = \langle A_i^T A_i, PP^T \rangle. \qquad \text{(FAIR-PCA)}$$

(We note that our FAIR-PCA is different from one in Samadi et al. [2018], but equivalent by additive and multiplicative scalings.) More generally, let $\mathcal{P}_d$ denote the set of all $n \times d$ projection matrices $P$, i.e., matrices with $d$ orthonormal columns. For each group $A_i$, we associate a function $f_i : \mathcal{P}_d \to \mathbb{R}$ that denotes the group's objective value for a particular projection. For any $g : \mathbb{R}^k \to \mathbb{R}$, we define the $(f, g)$-multi-criteria dimensionality reduction problem as finding a $d$-dimensional projection $P$ which optimizes

$$\max_{P \in \mathcal{P}_d} g(f_1(P), f_2(P), \ldots, f_k(P)). \qquad \text{(MULTI-CRITERIA-DIMENSION-REDUCTION)}$$

In the above example of max-min Fair-PCA, $g$ is simply the min function and $f_i(P) = \|A_i P\|^2$ is the total squared norm of the projection of vectors in $A_i$. Other examples include: defining each $f_i$ as the average squared norm of the projections rather than the total, or the marginal variance — the difference in total squared norm when using $P$ rather than the best possible projection for that group. One could also choose the product function $g(y_1, \ldots, y_k) = \prod_i y_i$ for the accumulating function $g$. This is also a natural choice, famously introduced in Nash's solution to the bargaining problemNash Jr [1950]; Kaneko and Nakamura [1979]. This framework can also describe the $p$th power mean of the projections, e.g. $f_i(P) = \|A_i P\|^2$ and $g(y_1, \ldots, y_k) = \left( \sum_{i\in[k]} y_i^{p/2} \right)^{1/p}$.

The appropriate weighting of $k$ objectives often depends on the context and application. The central motivating questions of this paper are the following:

$\diamond$ *What is the complexity of* FAIR-PCA *?*

$\diamond$ *More generally, what is the complexity of* MULTI-CRITERIA-DIMENSION-REDUCTION *?*

Framed another way, we ask whether these multi-criteria optimization problems force us to incur substantial computational cost compared to optimizing $g$ over $A$ alone. Samadi et al. [2018] introduced the problem of FAIR-PCA and showed how to use the natural semi-definite relaxation to find a rank-$(d + k - 1)$ approximation whose cost is at most that of the optimal rank-$d$ approximation. For $k = 2$ groups, this is an increase of 1 in the dimension (as opposed to the naïve bound of $2d$, by taking the span of the optimal $d$-dimensional subspaces for the two groups). The computational complexity of finding the exact optimal solution to FAIR-PCA was left as an open question.

## 1.1 Results and techniques

Let us first focus on FAIR-PCA for ease of exposition. The problem can be reformulated as the following mathematical program where we denote $PP^T$ by $X$. A natural approach to solving this problem is to consider the SDP relaxation obtained by relaxing the rank constraint to a bound on the trace.

| **Exact FAIR-PCA** | **SDP Relaxation of FAIR-PCA** |
|---|---|
| $\max\ z$ | $\max\ z$ |
| $\langle A_i^T A_i, X \rangle \geq z \quad i \in \{1, \ldots, k\}$ | $\langle A_i^T A_i, X \rangle \geq z \quad i \in \{1, \ldots, k\}$ |
| $\mathrm{rank}(X) \leq d$ | $\mathrm{tr}(X) \leq d$ |
| $0 \preceq X \preceq I$ | $0 \preceq X \preceq I$ |

Our first main result is that the SDP relaxation is exact when there are *two* groups. Thus finding an extreme point of this SDP gives an exact algorithm for FAIR-PCA for two groups. Previously, only approximation algorithms were known for this problem. This result also resolves the open problem posed by Samadi et al. [2018].

**Theorem 1.1.** *Any optimal extreme point solution to the SDP relaxation for* FAIR-PCA *with two groups has rank at most $d$. Therefore, 2-group* FAIR-PCA *can be solved in polynomial time.*

Given $m$ datapoints partitioned into $k \leq n$ groups in $n$ dimensions, the algorithm runs in $O(nm + n^{6.5})$ time. $O(mnk)$ is from computing $A_i^T A_i$ and $O(n^{6.5})$ is from solving an SDP over $n \times n$ PSD matrices [Ben-Tal and Nemirovski, 2001]. Our results also hold for the MULTI-CRITERIA-DIMENSION-REDUCTION when $g$ is monotone nondecreasing in any one coordinate and concave, and each $f_i$ is an affine function of $PP^T$ (and thus a special case of a quadratic function in $P$).

**Theorem 1.2.** *There is a polynomial time algorithm for 2-group* MULTI-CRITERIA-DIMENSION-REDUCTION *problem when $g$ is concave and monotone nondecreasing for at least one of its two arguments, and each $f_i$ is linear in $PP^T$, i.e., $f_i(P) = \langle B_i, PP^T \rangle$ for some matrix $B_i(A)$.*

As indicated in the theorem, the core idea is that extreme-point solutions of the SDP, in fact, have rank $d$, not just trace equal to $d$.

For $k > 2$, the SDP need not recover a rank $d$ solution. In fact, the SDP may be inexact even for $k = 3$ (see Section 8). Nonetheless, we show that we can bound the rank of a solution to the SDP and obtain the following result. We state it for FAIR-PCA, though the same bound holds for MULTI-CRITERIA-DIMENSION-REDUCTION under the same assumptions as in Theorem 1.1. Note that this result generalizes Theorem 1.1.

**Theorem 1.3.** *For any concave $g$ that is monotone nondecreasing in at least one of its arguments, there exists a polynomial time algorithm for* FAIR-PCA *with $k$ groups that returns a*

$d + \left\lfloor \sqrt{2k + \frac{1}{4}} - \frac{3}{2} \right\rfloor$-*dimensional embedding whose objective value is at least that of the optimal* $d$-*dimensional embedding. If $g$ is only concave, then the solution lies in at most $d + 1$ dimensions.*

This strictly improves and generalizes the bound of $d + k - 1$ for FAIR-PCA . Moreover, if the dimensionality of the solution is a hard constraint, instead of tolerating $s = O(\sqrt{k})$ extra dimension in the solution, one may solve FAIR-PCA for target dimension $d - s$ to guarantee a solution of rank at most $d$. Thus, we obtain an approximation algorithm for FAIR-PCA of factor $1 - \frac{O(\sqrt{k})}{d}$.

**Theorem 1.4.** *Let $A_1, \ldots, A_k$ be data sets of $k$ groups and suppose $s := \left\lfloor \sqrt{2k + \frac{1}{4}} - \frac{3}{2} \right\rfloor < d$. Then, there exists a polynomial-time approximation algorithm of factor $1 - \frac{s}{d} = 1 - \frac{O(\sqrt{k})}{d}$ to* FAIR-PCA *problem.*

That is, the algorithm returns a project $P \in \mathcal{P}_d$ of *exact* rank $d$ with objective at least $1 - \frac{s}{d}$ of the optimal objective. More details on the approximation result are in Section 4. The runtime of Theorems 1.2 and 1.3 depends on access to first order oracle to $g$ and standard application of the ellipsoid algorithm would take $\tilde{O}(n^2)$ oracle calls.

We now focus our attention to the marginal loss function. This measures the maximum over the groups of the difference between the variance of a common solution for the $k$ groups and an optimal solution for an individual group ("the marginal cost of sharing a common subspace"). For this problem, the above scaling method could substantially harm the objective value, since the target function is nonlinear. MULTI-CRITERIA-DIMENSION-REDUCTION captures the marginal loss functions by setting the utility $f_i(P) = \|A_i P\|_F^2 - \max_{Q \in \mathcal{P}_d} \|A_i Q\|_F^2$ for each group $i$ and $g(f_1, f_2, \ldots, f_k) := \min\{f_1, f_2, \ldots, f_k\}$, giving an optimization problem

$$\min_{P \in \mathcal{P}_d} \max_{i \in [k]} \left( \max_{Q \in \mathcal{P}_d} \|A_i Q\|_F^2 - \|A_i P\|_F^2 \right) \tag{1}$$

and the marginal loss objective is indeed the objective of the problem.

In Section 5, we develop a general rounding framework for SDPs with eigenvalue upper bounds and $k$ other linear constraints. This algorithm gives a solution of desired rank that violates each constraint by a bounded amount. The precise statement is Theorem 1.8. It implies that for FAIR-PCA with marginal loss as the objective the additive error is

$$\Delta(\mathcal{A}) := \max_{S \subseteq [m]} \sum_{i=1}^{\lfloor \sqrt{2|S|+1} \rfloor} \sigma_i(A_S)$$

where $A_S = \frac{1}{|S|} \sum_{i \in S} A_i$.

It is natural to ask whether FAIR-PCA is NP-hard to solve exactly. The following result implies that it is, even for the target dimension $d = 1$.

**Theorem 1.5.** *The max-min* FAIR-PCA *problem for target dimension $d = 1$ is NP-hard when the number of groups $k$ is part of the input.*

This raises the question of the complexity for constant $k \geq 3$ groups. For $k$ groups, we would have $k$ constraints, one for each group, plus the eigenvalue constraint and the trace constraint; now the tractability of the problem is far from clear. In fact, as we show in Section 8, the SDP has an integrality gap even for $k = 3, d = 1$. We therefore consider an approach beyond SDPs, to one that involves solving non-convex problems. Thanks to the powerful algorithmic theory of quadratic maps, developed by Grigoriev and Pasechnik [2005], it is polynomial-time solvable to check feasibility of a set of quadratic constraints for any fixed $k$. As we discuss next, their algorithm can check for zeros of a function of a set of $k$ quadratic functions, and can be used to optimize the function. Using this result, we show that for $d = k = O(1)$, there is a polynomial-time algorithm for rather general functions $g$ of the values of individual groups.

**Theorem 1.6.** *Let the fairness objective be $g : \mathbb{R}^k \to \mathbb{R}$ where $g$ is a degree $\ell$ polynomial in some computable subring of $\mathbb{R}^k$ and each $f_i$ is quadratic for $1 \leq i \leq k$. Then there is an algorithm to solve the fair dimensionality reduction problem in time $(\ell d n)^{O(k+d^2)}$.*

By choosing $g$ to be the product polynomial over the usual $(\times, +)$ ring or the $\min$ function which is degree $k$ in the $(\min, +)$ ring, this applies to the variants of FAIR-PCA discussed above and various other problems.

**SDP extreme points.** For $k = 2$, the underlying structural property we show is that extreme point solutions of the SDP have rank exactly $d$. First, for $k = d = 1$, this is the largest eigenvalue problem, since the maximum obtained by a matrix of trace equal to $1$ can also be obtained by one of the extreme points in the convex decomposition of this matrix. This extends to trace equal to any $d$, i.e., the optimal solution must be given by the top $k$ eigenvectors of $A^T A$. Second, without the eigenvalue bound, for any SDP with $k$ constraints, there is an upper bound on the rank of any extreme point, of $O(\sqrt{k})$, a seminal result of Pataki [1998] (see also Barvinok [1995]). However, we cannot apply this directly as we have the eigenvalue upper bound constraint. The complication here is that we have to take into account the constraint $X \preceq I$ without increasing the rank.

**Theorem 1.7.** *Let $C$ and $A_1, \ldots, A_m$ be $n \times n$ real matrices, $d \leq n$, and $b_1, \ldots b_m \in \mathbb{R}$. Suppose the semi-definite program $\mathbb{SDP}(\mathbb{I})$:*

$$\min \langle C, X \rangle \text{ subject to} \tag{2}$$
$$\langle A_i, X \rangle \quad \triangleleft_i \quad b_i \ \ \forall \, 1 \leq i \leq m \tag{3}$$
$$\operatorname{tr}(X) \quad \leq \quad d \tag{4}$$
$$0 \preceq X \quad \preceq \quad I_n \tag{5}$$

*where $\triangleleft_i \in \{\leq, \geq, =\}$, has a nonempty feasible set. Then, all extreme optimal solutions $X^*$ to $\mathbb{SDP}(\mathbb{I})$ have rank at most $r^* := d + \left\lfloor \sqrt{2m + \frac{9}{4}} - \frac{3}{2} \right\rfloor$. Moreover, given a feasible optimal solution, an extreme optimal solution can be found in polynomial time.*

To prove the theorem, we extend Pataki [1998]'s characterization of rank of SDP extreme points with minimal loss in the rank. We show that the constraints $0 \preceq X \preceq I$ can be interpreted as a generalization of restricting variables to lie between $0$ and $1$ in the case of linear programming relaxations. From a technical perspective, our results give new insights into structural properties of extreme points of semi-definite programs and more general convex programs. Since the result of Pataki [1998] has been studied from perspective of fast algorithms Boumal et al. [2016]; Burer and Monteiro [2003, 2005] and applied in community detection and phase synchronization Bandeira et al. [2016], we expect our extension of the result to have further applications in many of these areas.

**SDP iterative rounding.** Using Theorem 1.7, we extend the iterative rounding framework for linear programs (see Lau et al. [2011] and references therein) to semi-definite programs, where the $0, 1$ constraints are generalized to eigenvalue bounds. The algorithm has a remarkably similar flavor. In each iteration, we fix the subspaces spanned by eigenvectors with $0$ and $1$ eigenvalues, and argue that one of the constraints can be dropped while bounding the total violation in the constraint over the course of the algorithm. While this applies directly to the FAIR-PCA problem, in fact, is a general statement for SDPs, which we give below.

Let $\mathcal{A} = \{A_1, \ldots, A_m\}$ be a collection of $n \times n$ matrices. For any set $S \subseteq \{1, \ldots, m\}$, let $\sigma_i(S)$ the $i^{th}$ largest singular of the average of matrices $\frac{1}{|S|} \sum_{i \in S} A_i$. We let

$$\Delta(\mathcal{A}) := \max_{S \subseteq [m]} \sum_{i=1}^{\lfloor \sqrt{2|S|} + 1 \rfloor} \sigma_i(S).$$

**Theorem 1.8.** *Let $C$ be a $n \times n$ matrix and $\mathcal{A} = \{A_1, \ldots, A_m\}$ be a collection of $n \times n$ real matrices, $d \leq n$, and $b_1, \ldots b_m \in \mathbb{R}$. Suppose the semi-definite program $\mathbb{SDP}$:*

$$\min \langle C, X \rangle \text{ subject to}$$
$$\langle A_i, X \rangle \quad \geq \quad b_i \ \ \forall \, 1 \leq i \leq m$$
$$\operatorname{tr}(X) \quad \leq \quad d$$
$$0 \preceq X \quad \preceq \quad I_n$$

*has a nonempty feasible set and let $X^*$ denote an optimal solution. The Algorithm ITERATIVE-SDP (see Figure 2 in Appendix) returns a matrix $\tilde{X}$ such that*

1. *rank of $\tilde{X}$ is at most d,*

2. $\langle C, \tilde{X} \rangle \leq \langle C, X^* \rangle$, *and*

3. $\langle A_i, \tilde{X} \rangle \geq b_i - \Delta(\mathcal{A})$ *for each* $1 \leq i \leq m$.

The time complexity of Theorems 1.7 and 1.8 is analyzed in Sections 2 and 5. Both algorithms introduce the rounding procedures that do not contribute significant computational cost; rather, solving the SDPis the bottleneck for running time both in theory and practice.

## 1.2 Related work

As mentioned earlier, Pataki [1998] (see also Barvinok [1995]) showed low rank solutions to semi-definite programs with a small number of affine constraints can be obtained efficiently. Restricting a feasible region of certain SDPs relaxations with low-rank constraints has been shown to avoid spurious local optima [Bandeira et al., 2016] and reduce the runtime due to known heuristics and analysis [Burer and Monteiro, 2003, 2005; Boumal et al., 2016]. We also remark that methods based on Johnson-Lindenstrauss lemma can also be applied to obtain bi-criteria results for FAIR-PCA problem. For example, So et al. [2008] give algorithms that give low rank solutions for SDPs with affine constraints without the upper bound on eigenvalues. Here we have focused on single criteria setting, with violation either in the number of dimensions or the objective but not both. We also remark that extreme point solutions to linear programming have played an important role in the design of approximation algorithms [Lau et al., 2011] and our result adds to the comparatively small, but growing, number of applications for utilizing extreme points of semi-definite programs.

A closely related area, especially to MULTI-CRITERIA-DIMENSION-REDUCTION problem, is multi-objective optimization which has a vast literature. We refer the reader to Deb [2014] and references therein. We also remark that properties of extreme point solutions of linear programs [Ravi and Goemans, 1996; Grandoni et al., 2014] have also been utilized to obtain approximation algorithms to multi-objective problems. For semi-definite programming based methods, the closest works are on simultaneous max-cut [Bhangale et al., 2015, 2018] that utilize the sum of squares hierarchy to obtain improved approximation algorithms.

The applications of multi-criteria dimensionality reduction in fairness are closely related to studies on representational bias in machine learning [Crawford, 2017; Noble, 2018; Bolukbasi et al., 2016] and fair resource allocation in game theory [Wei et al., 2010; Fang and Bensaou, 2004]. There have been various mathematical formulations suggested for representational bias in ML [Chierichetti et al., 2017; Celis et al., 2018; Kleindessner et al., 2019; Samadi et al., 2018] among which our model covers unbalanced reconstruction error in PCA suggested by Samadi et al. [2018]. From the game theory literature, our model covers Nash social welfare objective [Kaneko and Nakamura, 1979; Nash Jr, 1950] and others [Kalai et al., 1975; Kalai, 1977].

## 2 Low-rank solutions of MULTI-CRITERIA-DIMENSION-REDUCTION

In this section, we show that all extreme solutions of SDP relaxation of MULTI-CRITERIA-DIMENSION-REDUCTION have low rank, proving Theorem 1.1-1.3. Before we state the results, we make the following assumptions. In this section, we let $g : \mathbb{R}^k \to \mathbb{R}$ be a concave function which is monotonic in at least one coordinate, and mildly assume that $g$ can be accessed with a polynomial-time subgradient oracle and is polynomially bounded by its input. We are explicitly given functions $f_1, f_2, \ldots, f_k$ which are affine in $PP^T$, i.e. we are given real $n \times n$ matrices $B_1, \ldots, B_k$ and constants $\alpha_1, \alpha_2, \ldots, \alpha_k \in \mathbb{R}$ and $f_i(P) = \langle B_i, PP^T \rangle + \alpha_i$.

We assume $g$ to be $G$-Lipschitz. For functions $f_1, \ldots, f_k, g$ that are $L_1, \ldots, L_k, G$-Lipschitz, we define an $\epsilon$-optimal solution to $(f, g)$-MULTI-CRITERIA-DIMENSION-REDUCTION problem as a projection matrix $X \in \mathbb{R}^{n \times n}, 0 \preceq X \preceq I_n$ of rank $d$ whose objective value is at most $G\epsilon \left( \sum_{i=1}^k L_i^2 \right)^{1/2}$ from the optimum. In the context where an optimization problem has affine constraints $F_i(X) \leq b_i$ where $F_i$ is $L_i$ Lipschitz, we also define $\epsilon$-solution as a projection matrix $X \in \mathbb{R}^{n \times n}, 0 \preceq X \preceq I_n$ of rank $d$ that violates $i$th affine constraints by at most $\epsilon L_i$. Note that the feasible region of the problem is implicitly bounded by the constraint $X \preceq I_n$.

In this section, the algorithm may involve solving an optimization under a matrix linear inequality, which may not give an answer representable in finite bits of computation. However, we give algorithms that return an $\epsilon$-close solution whose running time depends polynomially on $\log \frac{1}{\epsilon}$ for any $\epsilon > 0$. This is standard for computational tractability in convex optimization (see, for example, in Ben-Tal and Nemirovski [2001]). Therefore, for ease of exposition, we omit the computational error dependent on this $\epsilon$ to obtain an $\epsilon$-feasible and $\epsilon$-optimal solution, and define polynomial running time as polynomial in $n, k$ and $\log \frac{1}{\epsilon}$.

We first prove Theorem 1.7 below. To prove Theorem 1.1-1.3, we first show that extreme point solutions in semi-definite cone under affine constraints and $X \preceq I$ have low rank. The statement builds on a result of Pataki [1998]. We then apply our result to MULTI-CRITERIA-DIMENSION-REDUCTION problem, which contains the FAIR-PCA problem. Finally, we show that the existence of a low-rank solution leads to an approximation algorithm to FAIR-PCA problem.

**Proof of Theorem 1.7**: Let $X^*$ be an extreme point optimal solution to $\mathbb{SDP}(\mathbb{I})$. Suppose rank of $X^*$, say $r$, is more than $r^*$. Then we show a contradiction to the fact that $X^*$ is extreme. Let $0 \le l \le r$ of the eigenvalues of $X^*$ be equal to one. If $l \ge d$, then we have $l = r = d$ since $\text{tr}(X) \le d$ and we are done. Thus we assume that $l \le d - 1$. In that case, there exist matrices $Q_1 \in \mathbb{R}^{n \times r - l}$, $Q_2 \in \mathbb{R}^{n \times l}$ and a symmetric matrix $\Lambda \in \mathbb{R}^{(r-l) \times (r-l)}$ such that

$$X^* = (Q_1 \quad Q_2) \begin{pmatrix} \Lambda & 0 \\ 0 & I_l \end{pmatrix} (Q_1 \quad Q_2)^\top = Q_1 \Lambda Q_1^\top + Q_2 Q_2^T$$

where $0 \prec \Lambda \prec I_{r-l}$, $Q_1^T Q_1 = I_{r-l}$, $Q_2^T Q_2 = I_l$, and that the columns of $Q_1$ and $Q_2$ are orthogonal, i.e. $Q = (Q_1 \quad Q_2)$ has orthonormal columns. Now, we have

$$\langle A_i, X^* \rangle = \langle A_i, Q_1 \Lambda Q_1^\top + Q_2 Q_2^\top \rangle = \langle Q_1^\top A_i Q_1, \Lambda \rangle + \langle A_i, Q_2 Q_2^\top \rangle$$

and $\text{tr}(X^*) = \langle Q_1^\top Q_1, \Lambda \rangle + \text{tr}(Q_2 Q_2^\top)$ so that $\langle A_i, X^* \rangle$ and $\text{tr}(X^*)$ are linear in $\Lambda$.

Observe the set of $s \times s$ symmetric matrices forms a vector space of dimension $\frac{s(s+1)}{2}$ with the above inner product where we consider the matrices as long vectors. If $m + 1 < \frac{(r-l)(r-l+1)}{2}$ then there exists a $(r - l) \times (r - l)$-symmetric matrix $\Delta \ne 0$ such that $\langle Q_1^\top A_i Q_1, \Delta \rangle = 0$ for each $1 \le i \le m$ and $\langle Q_1^\top Q_1, \Delta \rangle = 0$.

But then we claim that $Q_1(\Lambda \pm \delta\Delta)Q_1^\top + Q_2 Q_2^T$ is feasible for small $\delta > 0$, which implies a contradiction to $X^*$ being extreme. Indeed, it satisfies all the linear constraints by construction of $\Delta$. Thus it remains to check the eigenvalues of the newly constructed matrix. Observe that

$$Q_1(\Lambda \pm \delta\Delta)Q_1^\top + Q_2 Q_2^T = Q \begin{pmatrix} \Lambda \pm \delta\Delta & 0 \\ 0 & I_l \end{pmatrix} Q^\top$$

with orthonormal $Q$. Thus it is enough to consider the eigenvalues of $\begin{pmatrix} \Lambda \pm \delta\Delta & 0 \\ 0 & I_l \end{pmatrix}$.

Observe that eigenvalues of the above matrix are exactly $l$ ones and eigenvalues of $\Lambda \pm \delta\Delta$. Since eigenvalues of $\Lambda$ are bounded away from 0 and 1, one can find small $\delta$ such that the eigenvalues of $\Lambda \pm \delta\Delta$ are bounded away from 0 and 1 as well, so we are done. Therefore, we must have $m + 1 \ge \frac{(r-l)(r-l+1)}{2}$ which implies $r - l \le -\frac{1}{2} + \sqrt{2m + \frac{9}{4}}$. By $l \le d - 1$, we have $r \le r^*$.

For the algorithmic version, given feasible $\bar{X}$, we iteratively reduce $r - l$ by at least one until $m + 1 \ge \frac{(r-l)(r-l+1)}{2}$. While $m + 1 < \frac{(r-l)(r-l+1)}{2}$, we obtain $\Delta$ by using Gaussian elimination. Now we want to find the correct value of $\pm\delta$ so that $\Lambda' = \Lambda \pm \delta\Delta$ takes one of the eigenvalues to zero or one. First, determine the sign of $\langle C, \Delta \rangle$ to find the correct sign to move $\Lambda$ that keeps the objective non-increasing, say it is in the positive direction. Since the set of feasible $X$ is convex and bounded, the ray $f(t) = Q_1(\Lambda + t\Delta)Q_1^\top + Q_2 Q_2^\top, t \ge 0$ intersects the boundary of feasible region at a unique $t' > 0$. Perform binary search for the correct value of $t'$ and set $\delta = t'$ up to the desired accuracy. Since $\langle Q_1^\top A_i Q_1, \Delta \rangle = 0$ for each $1 \le i \le m$ and $\langle Q_1^\top Q_1, \Delta \rangle = 0$, the additional tight constraint from moving $\Lambda' \leftarrow \Lambda + \delta\Delta$ to the boundary of feasible region must be an eigenvalue constraint $0 \preceq X \preceq I_n$, i.e., at least one additional eigenvalue is now at 0 or 1, as desired. We apply eigenvalue decomposition to $\Lambda'$ and update $Q_1$ accordingly, and repeat.

The algorithm involves at most $n$ rounds of reducing $r - l$, each of which involves Gaussian elimination and several iterations (from binary search) of $0 \preceq X \preceq I_n$ which can be done by eigenvalue value decomposition. Gaussian elimination and eigenvalue decomposition can be done in $O(n^3)$ time, and therefore the total runtime of SDP rounding is $\tilde{O}(n^4)$ which is polynomial. $\square$

In practice, one may initially reduce the rank of given feasible $\bar{X}$ using an LP rounding (in $O(n^{3.5})$ time) introduced in Samadi et al. [2018] so that the number of rounds of reducing $r - l$ is further bounded by $k - 1$. The runtime complexity is then $O(n^{3.5}) + \tilde{O}(kn^3)$.

The next corollary is obtained from the bound $r - l \leq -\frac{1}{2} + \sqrt{2m + \frac{9}{4}}$ in the proof of Theorem 1.7.

**Corollary 2.1.** *The number of fractional eigenvalues in any extreme point solution $X$ to $\mathbb{SDP}(\mathbb{I})$ is bounded by $\sqrt{2m + \frac{9}{4}} - \frac{1}{2} \leq \lfloor \sqrt{2m} + 1 \rfloor$.*

We are now ready to state the main result of this section that we can find a low-rank solution for MULTI-CRITERIA-DIMENSION-REDUCTION . Recall that $\mathcal{P}_d$ is the set of all $n \times d$ projection matrices $P$, i.e., matrices with $d$ orthonormal columns and the $(f, g)$-MULTI-CRITERIA-DIMENSION-REDUCTION problem is to solve

$$\max_{P \in \mathcal{P}_d} g(f_1(P), f_2(P), \ldots, f_k(P)) \tag{6}$$

**Theorem 2.2.** *There exists a polynomial-time algorithm to solve $(f, g)$-MULTI-CRITERIA-DIMENSION-REDUCTION that returns a solution $\hat{X}$ of rank at most $r^* := d + \left\lfloor \sqrt{2k + \frac{1}{4}} - \frac{3}{2} \right\rfloor$ whose objective value is at least that of the optimal $d$-dimensional embedding.*

The proof of Theorem 2.2 appears in Appendix. If the assumption that $g$ is monotonic in at least one coordinate is dropped, Theorem 2.2 will hold with $r^*$ by indexing constraints (11) in $\mathbb{SDP}(\mathbb{II})$ for all groups instead of $k - 1$ groups.

## 3 Experiments

First, we note that experiments for two groups were done in Samadi et al. [2018]. The algorithm outputs optimal solutions with exact rank, despite their weaker guarantee that the rank may be violated by at most 1. Hence, our result of Theorem 1.1 is a mathematical explanation of their missing empirical finding for two groups. We extend their experiments to more number of groups and objectives as follows (See Appendix for results on NSW objective and an additional dataset).

We perform experiments using the algorithm as outlined in Section 2 on the Default Credit data set [Yeh and Lien, 2009] for different target dimensions $d$. The data is partitioned into $k = 4, 6$ groups by education and gender and preprocessed to have mean zero and the same variance over features. We specified our algorithms by two objectives for MULTI-CRITERIA-DIMENSION-REDUCTION problem introduced earlier: the marginal loss function and Nash social welfare. The code is publicly available at `https://github.com/SDPforAll/multiCriteriaDimReduction`. Figure 1 shows the marginal loss by our algorithms compared to the standard PCA. Our algorithms significantly reduce "unfairness" in terms of the marginal loss that the standard PCA introduces.

In the experiments, extreme point solutions from SDPs enjoy lower rank violation than our worst-case guarantee. Indeed, while the guarantee is that the numbers of additional rank are at most $s = 1, 2$ for $k = 4, 6$, almost all SDP solutions have *exact* rank, and in rare cases when the solutions are not exact, the rank violation is only one. While we know that our rank violation guarantee cannot be improved in general (due to the integrality gap in Section 8), this opens a question of whether the guarantee is better for instances that arise in practice.

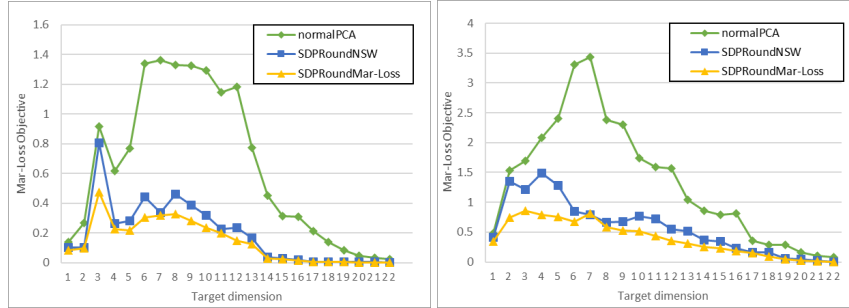

Figure 1: Marginal loss function (see (1)) of standard PCA compared to our SDP-based algorithms on Default Credit data. SDPRoundNSW and SDPRoundMar-Loss are two runs of the SDP-based algorithms maximizing NSW and minimizing marginal loss. Left: $k = 4$ groups. Right: $k = 6$.

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
