[Supplementary Material]

## Appendix

**Proof of Theorem 2.2**: First, we write a relaxation of (6):

$$\max_{X\in\mathbb{R}^{n\times n}} g(\langle B_1, X\rangle + \alpha_1, \ldots, \langle B_k, X\rangle + \alpha_k) \text{ subject to} \tag{7}$$

$$\operatorname{tr}(X) \leq d \tag{8}$$

$$0 \preceq X \preceq I_n \tag{9}$$

Since $g(x)$ is concave in $x \in \mathbb{R}^k$ and $\langle B_i, X\rangle + \alpha_i$ is affine in $X \in \mathbb{R}^{n\times n}$, we have that $g$ as a function of $X$ is also concave in $X$. By assumptions on $g$, and the fact that the feasible set is convex and bounded, we can solve the convex program in polynomial time, e.g. by ellipsoid method, to obtain a (possibly high-rank) optimal solution $\bar{X} \in \mathbb{R}^{n\times n}$. (In the case that $f_i$ is linear, the relaxation is also an SDP and may be solved faster in theory and practice). By assumptions on $g$, without loss of generality, we let $g$ be nondecreasing in the first coordinate. To reduce the rank of $\bar{X}$, we consider an $\mathbb{SDP}(\mathbb{II})$:

$$\max_{X\in\mathbb{R}^{n\times n}} \quad \langle B_1, X\rangle \text{ subject to} \tag{10}$$

$$\langle B_i, X\rangle = \langle B_i, \bar{X}\rangle \qquad \forall\, 2 \leq i \leq k \tag{11}$$

$$\operatorname{tr}(X) \leq d \tag{12}$$

$$0 \preceq X \preceq I_n \tag{13}$$

$\mathbb{SDP}(\mathbb{II})$ has a feasible solution $\bar{X}$ of objective $\langle B_1, X\rangle$ and note that there are $k-1$ constraints in (11). Hence, we can apply the algorithm in Theorem 1.7 with $m = k - 1$ to find an extreme solution $X^*$ of $\mathbb{SDP}(\mathbb{II})$ of rank at most $r^*$. Since $g$ is nondecreasing in $\langle B_1, X\rangle$, optimal solutions to $\mathbb{SDP}(\mathbb{II})$ gives objective value $g$ at least the optimum of the relaxation and hence at least the optimum of the original MULTI-CRITERIA-DIMENSION-REDUCTION problem. □

Another way to state Theorem 2.2 is that the number of groups must reach $\frac{(s+1)(s+2)}{2}$ before additional $s$ dimensions in the solution matrix $P$ is required to achieve the optimal objective value. For $k = 2$, no additional dimension in the solution is necessary to attain the optimum. We state this fact as follows. In particular, it applies to FAIR-PCA with two groups, proving Theorem 1.1.

**Corollary 3.1.** *The $(f, g)$-MULTI-CRITERIA-DIMENSION-REDUCTION problem on two groups can be solved in polynomial time.*

## 4 Approximation algorithm for FAIR-PCA

Recall that we require $s := \left\lfloor \sqrt{2k + \frac{1}{4}} - \frac{3}{2} \right\rfloor$ additional dimensions for the projection to achieve the optimal objective. One way to ensure that the algorithm outputs $d$-dimensional projection is to solve the problem in lower target dimension $d - s$, then apply the rounding described in Section 2. The relationship of objectives between problems with target dimension $d - s$ and $d$ is at most $\frac{d-s}{d}$ factor apart for FAIR-PCA problem because the objective scales linearly with $P$, giving an approximation guarantee of $1 - \frac{s}{d}$. Recall that given $A_1, \ldots, A_k$, FAIR-PCA problem is to solve

$$\max_{P: P^T P = I_d} \min_{1 \leq i \leq k} \|A_i P\|_F^2 = \langle A_i^T A_i, PP^T\rangle$$

We state the approximation guarantee and the algorithm formally as follows.

**Corollary 4.1.** *Let $A_1, \ldots, A_k$ be data sets of $k$ groups and suppose $s := \left\lfloor \sqrt{2k + \frac{1}{4}} - \frac{3}{2} \right\rfloor < d$.*

*Then there exists a polynomial-time approximation algorithm of factor $1 - \frac{s}{d} = 1 - \frac{O(\sqrt{k})}{d}$ to FAIR-PCA problem.*

*Proof.* We find an extreme solution $X^*$ of the FAIR-PCA problem of finding a projection from $n$ to $d - s$ target dimensions. By Theorem 2.2, the rank of $X^*$ is at most $d$.

Denote $\text{OPT}_d, X_d^*$ the optimal value and an optimal solution to FAIR-PCA with target dimension $d$. Note that $\frac{d-s}{d} X_d^*$ is a feasible solution to FAIR-PCA relaxation on target dimension $d - s$ which is at least $\frac{d-s}{d}\text{OPT}_d$ because the objective scales linearly with $X$. Therefore, the optimal FAIR-PCA relaxation of target dimension $d - s$ attains optimum at least $\frac{d-s}{d}\text{OPT}_d$, giving $(1 - \frac{s}{d})$-approximation ratio. □

# 5 Iterative rounding framework with applications to FAIR-PCA

In this section, we first prove Theorem 1.8.

We give an iterative rounding algorithm. The algorithm maintains three subspaces that are mutually orthogonal. Let $F_0, F_1, F$ denote matrices whose columns form an orthonormal basis of these subspaces. We will also abuse notation and denote these matrices by sets of vectors in their columns. We let the rank of $F_0, F_1$ and $F$ be $r_0, r_1$ and $r$, respectively. We will ensure that $r_0 + r_1 + r = n$, i.e., vectors in $F_0, F_1$ and $F$ span $\mathbb{R}^n$.

We initialize $F_0 = F_1 = \emptyset$ and $F = I_n$. Over iterations, we increase the subspaces spanned by columns of $F_0$ and $F_1$ and decrease $F$ while maintaining pairwise orthogonality. The vectors in columns of $F_1$ will be eigenvectors of our final solution with an eigenvalue of 1. In each iteration, we project the constraint matrices $A_i$ orthogonal to $F_1$ and $F_0$. We will then formulate a residual SDP using columns of $F$ as a basis and thus the newly constructed matrices will have size $r \times r$. To readers familiar with the iterative rounding framework in linear programming, this generalizes the method of fixing certain variables to 0 or 1 and then formulating the residual problem. We also maintain a subset of constraints indexed by $S$ where $S$ is initialized to $\{1, \ldots, m\}$.

The algorithm is specified in Figure 2. In each iteration, we formulate the following $\mathbb{SDP}(r)$ with variables $X(r)$ which will be a $r \times r$ symmetric matrix. Recall $r$ is the number of columns in $F$.

$$\max \langle F^T C F, X(r) \rangle$$
$$\langle F^T A_i F, X(r) \rangle \geq b_i - F_1^T A_i F_1 \quad i \in S$$
$$\operatorname{tr}(X) \leq d - \operatorname{rank}(F_1)$$
$$0 \preceq X(r) \preceq I_r$$

---

1. Initialize $F_0, F_1$ to be empty matrices and $F = I_n$, $S \leftarrow \{1, \ldots, m\}$.
2. If the $\mathbb{SDP}$ is infeasible, declare infeasibility. Else,
3. While $F$ is not the empty matrix.
    (a) Solve $\mathbb{SDP}(r)$ to obtain extreme point $X^*(r) = \sum_{j=1}^r \lambda_j v_j v_j^T$ where $\lambda_j$ are the eigenvalues and $v_j \in \mathbb{R}^r$ are the corresponding eigenvectors.
    (b) For any eigenvector $v$ of $X^*(r)$ with eigenvalue 0, let $F_0 \leftarrow F_0 \cup \{Fv\}$.
    (c) For any eigenvector $v$ of $X^*(r)$ with eigenvalue 1, let $F_1 \leftarrow F_1 \cup \{Fv\}$.
    (d) Let $X_f = \sum_{j:0 < \lambda_j < 1} \lambda_j v_j v_j^T$. If there exists a constraint $i \in S$ such that $\langle F^T A_i F, X_f \rangle < \Delta(\mathcal{A})$, then $S \leftarrow S \setminus \{i\}$.
    (e) For every eigenvector $v$ of $X^*(r)$ with eigenvalue not equal to 0 or 1, consider the vectors $Fv$ and form a matrix with these columns and use it as the new $F$.
4. Return $\tilde{X} = F_1 F_1^T$.

---

Figure 2: Iterative Rounding Algorithm ITERATIVE-SDP.

It is easy to see that the semi-definite program remains feasible over all iterations if $\mathbb{SDP}$ is declared feasible in the first iteration. Indeed the solution $X_f$ defined at the end of any iteration is a feasible solution to the next iteration. We also need the following standard claim.

**Claim 5.1.** *Let $Y$ be a positive semi-definite matrix such that $Y \preceq I$ with $\operatorname{tr}(Y) \leq l$. Let $B$ be real matrix of the same size as $Y$ and let $\lambda_i(B)$ denote the $i^{th}$ largest singular value of $B$. Then*

$$\langle B, Y \rangle \leq \sum_{i=1}^l \lambda_i(B).$$

The following result follows from Corollary 2.1 and Claim 5.1. Recall that

$$\Delta(\mathcal{A}) := \max_{S \subseteq [m]} \sum_{i=1}^{\lfloor \sqrt{2|S|} + 1 \rfloor} \sigma_i(S).$$

where $\sigma_i(S)$ is the $i$'th largest singular value of $\frac{1}{|S|} \sum_{i \in S} A_i$.

We let $\Delta$ denote $\Delta(\mathcal{A})$ for the rest of the section.

**Lemma 5.2.** *Consider any extreme point solution $X(r)$ of $\mathbb{SDP}(r)$ such that $\mathrm{rank}(X(r)) > \mathrm{tr}(X(r))$. Let $X(r) = \sum_{j=1}^{r} \lambda_j v_j v_j^T$ be its eigenvalue decomposition and $X_f = \sum_{0 < \lambda_j < 1} \lambda_j v_j v_j^T$. Then there exists a constraint $i$ such that $\langle F^T A_i F, X_f \rangle < \Delta$.*

*Proof.* Let $l = |S|$. From Corollary 2.1, it follows that number of fractional eigenvalues of $X(r)$ is at most $-\frac{1}{2} + \sqrt{2l + \frac{9}{4}} \leq \sqrt{2l} + 1$. Observe that $l > 0$ since $\mathrm{rank}(X(r)) > \mathrm{tr}(X(r))$. Thus $\mathrm{rank}(X_f) \leq \sqrt{2l} + 1$. Moreover, $0 \preceq X_f \preceq I$, thus from Claim 5.1, we obtain that

$$\left\langle \sum_{j \in S} F^T A_j F, X_f \right\rangle \leq \sum_{i=1}^{\lfloor \sqrt{2l} + 1 \rfloor} \sigma_i \left( \sum_{j \in S} F^T A_j F \right) \leq \sum_{i=1}^{\lfloor \sqrt{2l} + 1 \rfloor} \sigma_i \left( \sum_{j \in S} A_j \right) \leq l \cdot \Delta$$

where the first inequality follows from Claim 5.1 and second inequality follows since the sum of top $l$ singular values reduces after projection. But then we obtain, by averaging, that there exists $j \in S$ such that

$$\langle F^T A_j F, X_f \rangle < \frac{1}{l} \cdot l\Delta = \Delta$$

as claimed. $\square$

Now we complete the proof of Theorem 1.8. Observe that the algorithm always maintains that end of each iteration, the trace of $X_f$ plus the rank of $F_1$ is at most $d$. Thus at the end of the algorithm, the returned solution has rank at most $d$. Next, consider the solution $X = F_1 F_1^T + F X_f F^T$ over the course of the algorithm. Again, it is easy to see that the objective value is non-increasing over the iterations. This follows since $X_f$ defined at the end of an iteration is a feasible solution to the next iteration.

Now we argue the violation in any constraint $i$. While the constraint $i$ remains in the SDP, the solution $X = F_1 F_1^T + F X_f F^T$ satisfies

$$\langle A_i, X \rangle = \langle A_i, F_1 F_1^T \rangle + \langle A_i, F X_f F^T \rangle$$
$$= \langle A_i, F_1 F_1^T \rangle + \langle F^T A_i F, X_f \rangle \leq \langle A_i, F_1 F_1^T \rangle + b_i - \langle A_i, F_1 F_1^T \rangle = b_i.$$

where the inequality again follows since $X_f$ is feasible with the updated constraints.

When constraint $i$ is removed it might be violated by a later solution. At this iteration, $\langle F^T A_i F, X_f \rangle \leq \Delta$. Thus, $\langle A_i, F_1 F_1^T \rangle \geq b_i - \Delta$. In the final solution, this bound can only go up as $F_1$ might only become larger. This completes the proof of the theorem.

We now analyze the runtime of the algorithm which contains at most $k$ iterations. Each iteration requires solving an SDP and eigenvector decompositions over $r \times r$ matrices and recomputing $F$. The SDP has runtime $O(r^{6.5})$ which exceeds eigenvector decomposition and computing $X_f, F$ takes $O(n^2)$. However, the result in Section 2 shows that $r \leq \sqrt{2k}$, and hence the total runtime of iterative rounding is $O(k^{4.25} + kn^2)$.

**Application to FAIR-PCA .** For the FAIR-PCA problem, iterative rounding recovers a rank-$d$ solution whose variance goes down from the SDP solution by at most $\Delta(\{A_1^T A_1, \ldots, A_k^T A_k\})$. While this is no better than what we get by scaling (Corollary 4.1) for the max variance objective function, when we consider the marginal loss, i.e., the difference between the variance of the common $d$-dimensional solution and the best $d$-dimensional solution for each group, then iterative rounding can be much better. The scaling solution guarantee relies on the max-variance being a concave function and for the marginal loss, the loss for each group could go up proportional to the *largest* max variance (largest sum of top $k$ singular values over the groups). With iterative rounding applied to the SDP solution, the loss $\Delta$ is the sum of only $O(\sqrt{k})$ singular values of the average of some subset of data matrices, so it can be better by as much as a factor of $\sqrt{k}$.

# 6 Polynomial time algorithm for fixed number of groups

**Functions of quadratic maps.** We briefly summarize the approach of Grigoriev and Pasechnik [2005]. Let $f_1, \ldots, f_k : \mathbb{R}^n \to \mathbb{R}$ be real-valued quadratic functions in $n$ variables. Let $p : \mathbb{R}^k \to \mathbb{R}$ be a polynomial of degree $\ell$ over some subring of $\mathbb{R}^k$ (e.g., the usual $(\times, +)$ or $(+, \min)$) The problem is to find all roots of the polynomial $p(f_1(x), f_2(x), \ldots, f_k(x))$, i.e., the set

$$Z = \{x \,:\, p(f_1(x), f_2(x), \ldots, f_k(x)) = 0\}.$$

First note that the set of solutions above is in general not finite and is some manifold and highly non-convex. The key idea of Grigoriev and Paleshnik (see also Barvinok Barvinok [1993] for a similar idea applied to a special case) is to show that this set of solutions can be partitioned into a relatively small number of connected components such that there is an into map from these components to roots of a univariate polynomial of degree $(\ell n)^{O(k)}$; this, therefore, bounds the total number of components. The proof of this mapping is based on an explicit decomposition of space with the property that if a piece of the decomposition has a solution, it must be the solution of a linear system. The number of possible such linear systems is bounded as $n^{O(k)}$, and these systems can be enumerated efficiently.

The core idea of the decomposition starts with the following simple observation that relies crucially on the maps being quadratic (and not of higher degree).

**Proposition 6.1.** *The partial derivatives of any degree $d$ polynomial $p$ of quadratic forms $f_i(x)$, where $f_i : \mathbb{R}^n \to \mathbb{R}$, is linear in $x$ for any fixed value of $\{f_1(x), \ldots, f_k(x)\}$.*

To see this, suppose $Y_j = f_j(x)$ and write

$$\frac{\partial p}{\partial x_i} = \sum_{j=1}^{k} \frac{\partial p(Y_1, \ldots, Y_k)}{\partial Y_j} \frac{\partial Y_j}{\partial x_i} = \sum_{j=1}^{k} \frac{\partial p(Y_1, \ldots, Y_k)}{\partial Y_j} \frac{\partial f_j(x)}{\partial x_i}.$$

Now the derivatives of $f_j$ are linear in $x_i$ as $f_j$ is quadratic, and so for any fixed values of $Y_1, \ldots, Y_k$, the expression is linear in $x$.

The next step is a nontrivial fact about connected components of analytic manifolds that holds in much greater generality. Instead of all points that correspond to zeros of $p$, we look at all "critical" points of $p$ defined as the set of points $x$ for which the partial derivatives in all but the first coordinate, i.e.,

$$Z_c = \{x \,:\, \frac{\partial p}{\partial x_i} = 0, \quad \forall 2 \le i \le n\}.$$

The theorem says that $Z_c$ will intersect every connected component of $Z$ [Grigor'ev and Vorobjov Jr, 1988].

Now the above two ideas can be combined as follows. We will cover all connected components of $Z_c$. To do this we consider, for each fixed value of $Y_1, \ldots, Y_k$, the possible solutions to the linear system obtained, alongside minimizing $x_1$. The rank of this system is in general at least $n - k$ after a small perturbation (while Grigoriev and Pasechnik [2005] uses a deterministic perturbation that takes some care, we could also use a small random perturbation). So the number of possible solutions grows only as exponential in $O(k)$ (and not $n$) and can be effectively enumerated in time $(\ell d)^{O(k)}$. This last step is highly nontrivial and needs the argument that over the reals, zeros from distinct components need only to be computed up to finite polynomial precision (as rationals) to keep them distinct. Thus, the perturbed version still covers all components of the original version. In this enumeration, we check for true solutions. The method actually works for any level set of $p$, $\{x \,:\, p(x) = t\}$ and not just its zeros. With this, we can optimize over $p$ as well. We conclude this section by paraphrasing the main theorem from Grigoriev and Pasechnik [2005].

**Theorem 6.2.** *[Grigoriev and Pasechnik, 2005] Given $k$ quadratic maps $q_1, \ldots, q_k : \mathbb{R}^k \to \mathbb{R}$ and a polynomial $p : \mathbb{R}^k \to \mathbb{R}$ over some computable subring of $\mathbb{R}$ of degree at most $\ell$, there is an algorithm to compute a set of points satisfying $p(q_1(x), \ldots, q_k(x)) = 0$ that meets each connected component of the set of zeros of $p$ using at most $(\ell n)^{O(k)}$ operations with all intermediate representations bounded by $(\ell n)^{O(k)}$ times the bit sizes of the coefficients of $p, q_1, \ldots, q_k$. The minimizer, maximizer or infimum of any polynomial $r(q_1(x), \ldots, q_k(x))$ of degree at most $\ell$ over the zeros of $p$ can also be computed in the same complexity.*

## 6.1 Proof of Theorem 1.6

We apply Theorem 6.2 and the corresponding algorithm as follows. Our variables will be the entries of an $n \times d$ matrix $P$. The quadratic maps will be $f_i(P)$ plus additional maps for $q_{ii}(P) = \|P_i\|^2 - 1$ and $q_{ij}(P) = P_i^T P_j$ for columns $P_i, P_j$ of $P$. The final polynomial is

$$p(f_1, \ldots, f_k, q_{11}, \ldots, q_{dd}) = \sum_{i \leq j} q_{ij}(P)^2.$$

We will find the maximum of the polynomial $r(f_1, \ldots f_k) = g(f_1, \ldots, f_k)$ over the set of zeros of $p$ using the algorithm of Theorem 6.2. Since the total number of variables is $dn$ and the number of quadratic maps is $k + d(d+1)/2$, we get the claimed complexity of $O(\ell dn)^{O(k+d^2)}$ operations and this times the input bit sizes as the bit complexity of the algorithm.

# 7 Hardness

**Theorem 7.1.** *The* FAIR-PCA *problem:*

$$\max_{z \in \mathbb{R}, P \in \mathbb{R}^{n \times d}} z \qquad \text{subject to} \tag{14}$$

$$\langle B_i, PP^T \rangle \geq z \qquad , \forall i \in [k] \tag{15}$$

$$P^T P = I_d \tag{16}$$

*for arbitrary $n \times n$ symmetric real PSD matrices $B_1, \ldots, B_k$ is NP-hard for $d = 1$ and $k = O(n)$.*

**Proof of Theorem 7.1**: We reduce another NP-hard problem of MAX-CUT to the stated fair PCA problem. In MAX-CUT, given a simple graph $G = (V, E)$, we optimize

$$\max_{S \subseteq V} e(S, V \setminus S) \tag{17}$$

over all subset $S$ of vertices. Here, $e(S, V \setminus S) = |\{e_{ij} \in E : i \in S, j \in V \setminus S\}|$ is the size of the cut $S$ in $G$. As common NP-hard problems, the decision version of MAX-CUT:

$$\exists? S \subseteq V : e(S, V \setminus S) \geq b \tag{18}$$

for an arbitrary $b > 0$ is also NP-hard. We may write MAX-CUT as an integer program as follows:

$$\exists? v \in \{-1, 1\}^V : \frac{1}{2} \sum_{ij \in E} (1 - v_i v_j) \geq b \tag{19}$$

Here $v_i$ represents whether a vertex $i$ is in the set $S$ or not:

$$v_i = \begin{cases} 1 & i \in S \\ -1 & i \notin S \end{cases} \tag{20}$$

and it can be easily verified that the objective represents the desired cut function.

We now show that this MAX-CUT integer feasibility problem can be formulated as an instance of the fair PCA problem (14)-(16). In fact, it will be formulated as a feasibility version of the fair PCA by checking if the optimal $z$ of an instance is at least $b$. We choose $d = 1$ and $n = |V|$ for this instance, and we write $P = [u_1; \ldots; u_n] \in \mathbb{R}^n$. The rest of the proof is to show that it is possible to construct constraints in the fair PCA form (15)-(16) to 1) enforce a discrete condition on $u_i$ to take only two values, behaving similarly as $v_i$; and 2) check an objective value of MAX-CUT.

The reason $u_i$ as written cannot behave exactly as $v_i$ is that constraint (16) requires $\sum_{i=1}^n u_i^2 = 1$ but $\sum_{i=1}^n v_i^2 = n$. Hence, we scale the variables in MAX-CUT problem by writing $v_i = \sqrt{n} u_i$ and rearrange terms in (19) to obtain an equivalent formulation of MAX-CUT:

$$\exists? u \in \{-\frac{1}{\sqrt{n}}, \frac{1}{\sqrt{n}}\}^n : n \sum_{ij \in E} -u_i u_j \geq 2b - |E| \tag{21}$$

We are now ready to give an explicit construction of $\{B_i\}_{i=1}^k$ to solve MAX-CUT formulation (21). Let $k = 2n + 1$. For each $j = 1, \ldots, n$, define

$$B_{2j-1} = bn \cdot \operatorname{diag}(\mathbf{e_j}), \ B_{2j} = \frac{bn}{n-1} \cdot \operatorname{diag}(\mathbf{1} - \mathbf{e_j})$$

where $\mathbf{e_j}$ and $\mathbf{1}$ denote vectors of length $n$ with all zeroes except one at the $j$th coordinate, and with all ones, respectively. It is clear that $B_{2j-1}, B_{2j}$ are PSD. Then for each $j = 1 \ldots, n$, the constraints $\langle B_{2j-1}, PP^T \rangle \geq b$ and $\langle B_{2j}, PP^T \rangle \geq b$ are equivalent to

$$u_j^2 \geq \frac{1}{n}, \ \text{and} \ \sum_{i \neq j} u_j^2 \geq \frac{n-1}{n}$$

respectively. Combining these two inequalities with $\sum_{i=1}^n u_i^2 = 1$ forces both inequalities to be equalities, implying that $u_j \in \{-\frac{1}{\sqrt{n}}, \frac{1}{\sqrt{n}}\}$ for all $j \in [n]$, as we aim.

Next, we set

$$B_{2n+1} = \frac{bn}{2b - |E| + n^2} \cdot (nI_n - A_G)$$

where $A_G = (\mathbb{I}[ij \in E])_{i,j \in [n]}$ is the adjacency matrix of the graph $G$. Since the matrix $nI_n - A_G$ is diagonally dominant and real symmetric, $B_{2n+1}$ is PSD. We have that $\langle B_{2n+1}, PP^T \rangle \geq b$ is equivalent to

$$\frac{bn}{2b - |E| + n^2} \left( n \sum_{i=1}^n u_i^2 - \sum_{ij \in E} u_i u_j \right) \geq b$$

which, by $\sum_{i=1}^n u_i^2 = 1$, is further equivalent to

$$n \sum_{ij \in E} -u_i u_j \geq 2b - |E|$$

To summarize, we constructed $B_1, \ldots, B_{2n+1}$ so that checking whether an objective of fair PCA is at least $b$ is equivalent to checking whether a graph $G$ has a cut of size at least $b$, which is NP-hard. □

## 8 Integrality gap

We showed that FAIR-PCA for $k = 2$ groups can be solved up to optimality in polynomial time using an SDP. For $k > 2$, we used a different, non-convex approach to get a polytime algorithm for any fixed $k, d$. Here we show that the SDP relaxation of FAIR-PCA has a gap even for $k = 3$ and $d = 1$.

**Lemma 8.1.** *The* FAIR-PCA *SDP relaxation:*

$$\begin{aligned}
\max \ &z \\
\langle B_i, X \rangle \geq z \quad &i \in \{1, \ldots, k\} \\
\operatorname{tr}(X) \leq d \\
0 \preceq X \preceq I
\end{aligned}$$

*for $k = 3$, $d = 1$, and arbitrary PSD $\{B_i\}_{i=1}^k$ contains a gap, i.e. the optimum value of the SDP relaxation is different from one of exact* FAIR-PCA *problem.*

**Proof of Lemma 8.1**: Let $B_1 = \begin{bmatrix} 2 & 1 \\ 1 & 1 \end{bmatrix}, B_2 = \begin{bmatrix} 1 & 1 \\ 1 & 2 \end{bmatrix}, B_3 = \begin{bmatrix} 2 & -1 \\ -1 & 2 \end{bmatrix}$. It can be checked that $B_i$ are PSD. The optimum of the relaxation is $7/4$ (given by the optimal solution $X = \begin{bmatrix} 1/2 & 1/8 \\ 1/8 & 1/2 \end{bmatrix}$).

However, an optimal exact FAIR-PCA solution is $\hat{X} = \begin{bmatrix} 16/17 & 4/17 \\ 4/17 & 1/17 \end{bmatrix}$ which gives an optimum $26/17$ (one way to solve for optimum rank-1 solution $\hat{X}$ is by parameterizing $\hat{X} = v(\theta)v(\theta)^T$ for $v(\theta) = [\cos \theta; \sin \theta], \theta \in [0, 2\pi)$). □

Figure 3: NSW objective of standard PCA compared to our SDP-based algorithms on Default Credit data. SDPRoundNSW and SDPRoundMar-Loss are two runs of the SDP algorithms maximizing NSW objective and minimizing maximum marginal loss. Left: $k = 4$ groups. Right: $k = 6$.

Figure 4: Marginal loss and NSW objective of standard PCA compared to our SDP-based algorithms on Adult Income data. SDPRoundNSW and SDPRoundMar-Loss are two runs of the SDP algorithms maximizing NSW objective and minimizing maximum marginal loss.

## 9 Extended experiments

We also assess the performance of PCA with NSW objective, summarized in Figure 3. For NSW, standard PCA performs marginally worse (about 10%) compared to our algorithms. It is worth noting from Figures 1 and 3 that our algorithms that try to optimize either marginal loss function or NSW also perform well on the other fairness objective, making these PCAs promising candidates for fairness application.

The same experiments were done on the Adult Income data [Repository]. Some categorial features are preprocessed into integers vectors and some categorical features and rows with missing values are discarded. The final preprocessed data contains $m = 32560$ data points in $n = 59$ dimensions, partitioned into $k = 5$ groups based on race. Figure 4 shows the performance of our SDP-based algorithms compared to standard PCA on marginal loss and NSW objectives. Similar to the Credit Data, optimizing either marginal loss or NSW gives a PCA solution that also performs well in another criterion, and better than the standard PCA in both objectives. Almost all SDP solutions are exact without any rank violation.

We found that the running time of solving SDP, which depends on $n$, is the bottleneck in all experiments. Each run (for one value of $d$) of the experiments is fast ($< 0.5$ seconds) on Default Credit data which has $n = 23$, whereas one on Adult Income data ($n = 59$) takes between 10 and 15 seconds. However, it is worth noting that the runtime does not increase in noticeably from the numbers of data points and groups: larger $m$ only increases the data preprocessing time to obtain $n \times n$ matrices and larger $k$ increases the number of constraints. SDP solver and rounding algorithms can handle a moderate number of affine constraints efficiently. This observation is as expected from the theoretical analysis.