[Reviews · NeurIPS 2019]

Reviewer 1



I found this paper difficult to read. Some of the reasons were structural - I found the main and readily usable contribution of this paper to be the actual algorithm( the multi-criteria dimensionality reduction algorithm). The problem is that this information is written in paragraph form (between line 259 to 269) instead of pseudo-code so that it is readily visible. I also did not like the naming and organisation of the sections. The authors used the chapter name 'results' to mean mathematical foundational results it might be more helpful to rename this chapter into something like 'theoretical background'. It might also be more useful to have this chapter after the related research chapter so that it becomes even more evident why that background theory is necessary and where is plugs into the current state of the art solutions. I am not familiar enough with the literature in this area to comment on originality of this work.

Reviewer 2



The paper makes interesting connection between fairness and multi-parameter optimization, bringing to bear powerful techniques (such as SDP relaxation) from the latter. It supports its theoretical findings with experimental validation whose results suggest that in practice the number of extra dimensions is even smaller than the analytical guarantees. The paper is notable for diversity of its tools (SDP relaxation, geometry of quadratic functions, reduction of MAX-CUT to Fair PCA, experimental results).

Reviewer 3



Originality: The authors’ solve a problem left open in a previous paper and made strictly improve on previous work for approximation algorithms. They do so by giving new insights into structural properties of extreme points of semi-definite programs and more general convex programs. As far as I understand, the algorithms presented in the paper are not substantially new, but the analysis of these algorithms is novel. Quality: The paper seems complete and the proofs appear correct. The paper tackles interesting problems and reaches satisfying conclusions. They essentially close the book on the k=2 case, make significant improvements for k>2 and leave open some questions for structured data. Clarity: The paper is well-written. The introduction is easy to follow. It does a good job introducing and motivating the problem. The contributions of the work are made obvious with clear theorem statements. I appreciated that the authors’ focus first on the simpler case of Fair-PCA. The use of this as a building block to understand the more general case is really helpful for the reader. The progression from Thm 1.1 to Thm 1.2 to Thm 1.3 made each result easier to understand. Significance: The problem they study, dimensionality reduction, is an important and common technique for data analysis. The authors’ provide convincing motivation for the multi-criteria variant of the problem. They focus on the fairness as a main motivation. The authors’ resolve an open problem. Samadi et al. introduced the problem of Fair-PCA and gave an approximation algorithm and left the computational complexity of finding the exact optimal solution as an open problem. The authors’ improve upon that approximation for all k (number of constraints) and solve the open problem for k=2. The previous interest in this problem suggest that it was be interesting to the community. The authors’ also make several contributions of independent interest. They give new insights into structural properties of extreme points of semi-definite programs and more general convex programs. This is not my area of expertise but the authors’ say that similar results have been used to study fast algorithms, community detection and phase synchronization so this does seem to indicate that these results may have applications elsewhere. They do seem like general enough results to be useful to other researchers. There are some avenues for future work. In their experiments indicate that the rank bounds presented in the paper (while better than those in previous papers) may be loose on real datasets. They prove in the appendix that it is not the case in general for k>2 that the SDP solution obtains the exact rank. However, this does seem to be the case in their experiments. This leaves open the question what what properties of the data would ensure this? Minor comments: - I thought the progression to Thm 1.4 was a bit harder to understand. It would be nice to have a formal statement of what you mean by an approximation algorithm for Fair-PCA. It’s sort of clear from the proceeding prose that it has to satisfy the rank condition but it would be nice to have this stated in the thm. - Similarly, you define the marginal loss function in words but it would be nice to have it defined in an equation. It would also be nice to have a reminder of this definition and some intuition for it because Figure 1. I found this figure a little confusing. - Integrality gap is a common term in the optimality community but may not be as common elsewhere. It would be nice to have a brief definition of it. - It would be nice to have pointers to where particularly proofs occur in the appendix. - Typo: line 183. Singular-> singular value - In line 186 it should be noted that Figure 2 is in the appendix.

[Author Response · NeurIPS 2019]

We thank the reviewers for the thoughtful review of our work. We would like to briefly clarify a few minor points and will happily incorporate the suggested improvements and changes to future drafts of this work.

There are four main contributions in the paper: 1) defining the multicriterial dimensionality reduction problem, and observing on real-world datasets the failure of standard techniques to adequately solve this problem, 2) designing and analyzing new algorithms to solve multicriteria dimensionality reduction 3) analyzing the complexity of solving the general multicriteria dimensionality reduction problem (for constant $k$ and then for general $k$) 4) empirically evaluating our algorithms.

The theoretical results are original, rather than background material known prior to this work. We will happily separate the relevant background content into a preliminaries section to make this distinction clear. We agree with R5 that extremal solutions to mathematical programs have found use in many contexts, and our analysis relies on this as well as new insights to the specific problem we formulate.

Specifically responding to R3, we agree that our work doesn't measure the 'downstream' effects of different amounts of reconstruction error; doing so seems interesting but domain-specific. We will spend some time thinking about an appropriate setting for measuring this.

Regarding R4's question about MW: we implemented MW to the same datasets and tested its performance and runtime. MW, in fact, runs very efficiently on both datasets (within few seconds) and scales even up to random instances of original dimensions 1000. The empirical objective value of MW produces optimal solutions in many cases. Moreover, we have a theoretical understanding as to why: in these real datasets, we find no degeneracy in SDP feasible set, and hence the solution is necessarily unique and extreme. This implementation is publicly available at the same link in the paper.

We note however some limitations to MW in this setting. The framework of MW in Samadi et al does not apply to as many objectives as SDPs. MW depends on solving linear constraints, and so the objective that cannot be obtained from applying min/max to linear functions cannot be solved by MW. For example, the marginal loss objective is solvable by MW, but not Nash social welfare. Also, one must tune the parameter of MW and hope that it will converge faster than the theoretical bounds (Samadi et al mentioned 10-15 steps on real-world datasets but theoretically the runtime bound is much looser).

[Meta-Review · NeurIPS 2019]

Recent work of Samadi et al. introduced the fair PCA problem where the goal is to find a projection that minimizes the error and furthermore the error is balanced across two groups in the population. The takeaway message from their paper was that adding one extra dimension is enough. Firstly, this paper resolves the main open question from the work of Samadi et al. by giving an algorithm that does not need to increase the dimension by one. Second, they push the framework fo fair PCA in some interesting new directions that allow for alternative notions of fairness and multiple groups. They improve the fairness penalty to be \sqrt{k} from k-1. The reviewers felt that the paper gives a compelling collection of results on an important topic.